# Associations of social determinants of health and patient safety in perinatal care: Protocol for a systematic review with meta-analysis

Katharina Averdunk[1]*, Céline Miani[2], Brigitte Strizek[3], Matthias Weigl[1]

1 Institute for Patient Safety (IfPS), Medical Faculty, University Hospital Bonn, Bonn, Germany,
2 Department of Epidemiology and International Public Health, School of Public Health, Bielefeld University, Bielefeld, Germany, 3 Department of Obstetrics and Prenatal Medicine, University Hospital Bonn, Bonn, Germany

* katharina.ahring-averdunk@ukbonn.de

## Abstract

### Background

Associations between adverse social conditions and poor health are well documented – also in perinatal care. However, research into the actual ramifications of such disparities for perinatal patient safety remains inconclusive. Therefore, to achieve a comprehensive understanding of the risk and burden of patient harm to disadvantaged people, we aim to systematically review current evidence on social determinants of health (SDoH) and perinatal patient safety.

### Objective

This study protocol outlines definitions, methods, and procedures for a systematic literature review with meta-analysis aiming to synthesise the research base on the associations between SDoH and patient safety in perinatal care.

### Methods

Adhering to PRISMA guidelines, a literature search will be conducted for a systematic review in MEDLINE (PubMed), Scopus database, CINAHL (EBSCO), and Embase (Elsevier) for quantitative studies reporting associations between SDoH and patient safety measures in perinatal care. Data extraction will include study design, population, SDoH variables, outcome measures, effect sizes, and control variables. If deemed feasible after assessment of heterogeneity, narrative synthesis of findings will be complemented by conducting meta-analyses of pooled effect sizes. Methodological quality of included studies will be assessed using JBI Critical Appraisal Tools, and the certainty of evidence using the GRADE tool. This protocol is registered on PROSPERO (CRD420251090149) and OSF (https://doi.org/10.17605/OSF.IO/UP3JS).

**Data availability statement:** No datasets were generated or analysed during the current study. All relevant data from this study will be made publicly available on Open Science Framework (OSF) upon study completion and publication.

**Funding:** The authors received no specific funding for the research. The publication was supported by the Open Access Publication Fund of the University of Bonn, Germany (www. open-access.uni-bonn.de/en/funding/fund-ing-articles). The funders had no role in study design, data collection and analysis, decision to publish, or preparation of the manuscript.

**Competing interests:** The authors have declared that no competing interests exist.

## Discussion

The review aligns well with current global efforts to promote safe perinatal care and presents an innovative, comprehensive approach for assessing the associations between SDoH and patient safety. The review will provide the first systematic synthesis of current evidence of SDoH and patient safety in perinatal care. Anticipated limitations include heterogeneity of study designs, measures, and outcomes, with expected predominance of observational studies, which may limit causal inferences. However, this review will provide a valuable foundation for further empirical research and interventions to enhance equitable and safe perinatal care.

## Introduction

Global initiatives, such as the WHO's Global Patient Safety Action Plan 2021–2030, emphasise the need to prioritise maternal and newborn health, and in particular the importance of integrating equity considerations into patient safety strategies [1]. Associations between adverse social conditions and poor health are well documented, even in high-income countries such as the UK and Germany [2,3]. These social determinants of health (SDoH) include individual and socio-economic factors, social and environmental context factors, and the accessibility of health services [4–7].

Within the field of perinatal care, international research has examined a range of adverse outcomes in the context of social deprivation. These outcomes encompass avoidable maternal and foetal mortality, preterm birth, low birth weight, and postpartum mental distress [8–20]. Concurrently, studies have suggested that specific SDoH may be associated not only with overall health outcomes but also with the safety of care [18,21,22]. Therefore, to understand the actual risk and burden of inequitable health and health care for disadvantaged patients in perinatal care, it is essential to investigate patient safety in this high-risk setting. Despite the extensive research conducted on SDoH in the context of perinatal outcomes, there is a paucity of compelling synthesised evidence regarding the sequelae of SDoH for patient safety.

Patient safety is concerned with the reduction of the 'risk of unnecessary harm associated with healthcare to an acceptable minimum' [23]. Patient safety measures include, but are not limited to, procedure-related outcomes such as safety incidents that 'resulted or could have resulted in patient harm' [23]. These encompass failures, critical events, or near-misses related to communication, patient management, or clinical procedures [24]. Safety can be assessed by employing objective measures such as patient safety indicators or triggers [25,26], or subjective measures, encompassing providers' or patients' experiences.

A comprehensive framework for assessing patient safety in perinatal care is not yet available, leading to variability in safety-related endpoints and limited ability to draw clear, evidence-based conclusions on perinatal patient safety. Notably, studies focusing on the impact of SDoH on perinatal outcomes tend to use isolated care and health outcome measurements such as severe morbidity and mortality

[8,12,13,19,27,28], preterm birth [9,12,13,16,28], or low birthweight [12,14,28]. Despite some overlap of these measures with obstetric safety indicators [25], or care quality outcome sets [29–31], they do not necessarily indicate whether poor patient safety occurred in individual cases. Although studies aimed to expand this focus by investigating maternal near-misses [32], the specific ramifications for patient safety currently remain ambiguous and inconclusive. Therefore, in order to establish a profound understanding of the intersection of SDoH and perinatal patient safety, it is imperative to aggregate and integrate extant evidence, with a further focus on comprehensive safety measurements [19].

To date, a systematic synthesis of evidence investigating associations between SDoH and perinatal patient safety is missing. Therefore, the forthcoming literature review will make a valuable contribution to advancing equity and safety in perinatal care by (1) expanding the understanding of methods used to investigate perinatal patient safety, (2) identifying specific risk constellations associated with poor safety among disadvantaged patients, and (3) highlighting key determinants and outcome variables to inform the development of future empirical studies.

## Objective

This study protocol has been developed for outlining methods and procedures for conducting a systematic literature review with meta-analysis. The reviews' objective is to synthesise and integrate the extant research base on the actual ramifications of social disparities for perinatal patient safety. Specifically, the review will be guided by the following research question: What evidence is available on the associations between SDoH and patient safety in perinatal care?

## Methods

In accordance with this protocol, a systematic literature review and meta-analysis will be conducted to determine the extant evidence on associations between various SDoH and patient safety in perinatal care. The study will adhere to PRISMA guidelines for reporting systematic reviews and meta-analyses [33]. For this study protocol, we used PRISMA-P guideline [34]. This protocol was designed prior to data collection, extraction and analysis. The study has not yet commenced and will be conducted from August 2025 to January 2026, including data collection and data synthesis. Upon completion, results will be made publicly available and published. The review study is registered on the International Prospective Register of Systematic Reviews (PROSPERO) database (CRD420251090149) and on Open Science Framework (OSF) Registries (https://doi.org/10.17605/OSF.IO/UP3JS).

## Eligibility

As recommended for reviews of associations that inherently lack an intervention as well as a comparison variable [35], studies will be selected according to the PEO (Population – Exposure – Outcome) criteria outlined below as well as study and report characteristics, respectively:

**Study designs.** We will include original studies reporting on associations between SDoH and patient safety. We will include quantitative observational studies, including retrospective and prospective, cross-sectional and longitudinal designs, and cohort and case-control studies. We will also include randomised controlled trials. We will exclude intervention studies, case reports, and qualitative studies, as they do not allow for extraction of quantitative association measures. We will include articles from peer-reviewed scientific journals from the year 2000 onwards.

**Population.** We aim to capture relevant cases throughout the perinatal period. Therefore, we will include patients during their pregnancy, childbirth and the puerperium, as well as their foetuses and neonates.

**Exposure.** Of interest are reported patient characteristics, captured in the concept of 'social determinants of health' (SDoH), defined by the WHO as 'the conditions in which people are born, grow, live, work and age' that 'have a powerful influence on health inequities' [4]. To operationalise SDoH, we will apply different proxy indicators, classified according to the Office of Disease Prevention and Health Promotion [6]: Economic stability, education access and quality, health care access and quality, neighbourhood and built environment, and social and community context. In absence of a

comprehensive complement of characteristics within each of these five domains, specification of determinants (and, consequently, search terms) will consider various academic and policy sources on SDoH as well as scientific experience. Additionally, we will incorporate individual constitutional factors, including age, gender, and ethnicity, as proposed in the Solar & Irwin Model [7], and also applied in recent research on perinatal quality and equity [29]. Lastly, in response to current global challenges and migratory movements, we will add 'refugee status' as an additional domain of SDoH, as introduced by Wenner, Razum, & Bozorgmehr [36]. We will also include studies that employ indices and other compound measures capturing SDoH. Since the purpose of this study is to focus on the social conditions that may affect perinatal safety, we will consequently exclude patients' health-related exposure conditions such as chronic or severe diseases, including mental health, substance abuse, and sexually transmitted diseases, and disabilities.

**Outcomes.** Of interest will be a broad set of outcomes pertaining to perinatal patient safety. We will thus include all empirical investigations that report on any type of quantifiable, subjective or objective safety outcome, as long the specific aim was to investigate perinatal patient safety. We will exclude studies that do not investigate patients' safety, i.e., patient or clinical outcomes without specific reference to safety or harm in context with care provision.

**Setting.** We will include studies conducted in any perinatal care setting, including antepartum, intrapartum, postpartum and neonatal care. We will exclude studies pertaining to paediatric cases after the neonatal period of four weeks postpartum. Setting characteristics that might affect comparability (e.g., country's income-level and overall safety situation, inpatient or outpatient care setting) will be documented for potential sub-group analyses.

## Search strategy

We will search the following electronic databases: MEDLINE (via PubMed), CINAHL (via EBSCO), Embase (via Elsevier), and Scopus database. Additionally, we will apply forward and backward citation tracing and snowballing techniques among similar systematic reviews [37]. Literature will be searched in MEDLINE (via PubMed) using medical subject headings (MeSH) and text words related to the population, exposure, and outcomes of interest (Table 1). These three main constructs will be linked using Boolean operator AND, while MeSH terms and text words within a main construct will be linked using OR. Search terms will be adapted to the different databases via Polyglot [38]. Finally, the search will be updated towards the end of the review process.

## Study selection

A step-wise, consecutive screening and selection procedure will be performed: Two independent and trained reviewers will conduct level 1 (title/abstract) and level 2 (full text) screening of literature search results, using Rayyan [39]. They will review eligibility for data pooling and meta-analysis, for both using a standardised form informed by our eligibility criteria. We will resolve disagreement through discussion, involving a third author, if appropriate.

## Data extraction

We will extract data from eligible studies, using a standardised form informed by our predefined study characteristics as well as population, exposure, outcome data of interest [40]. All procedures will be piloted in a subset of studies.

**Study design.** We will extract information on study design, location (country), health care setting (inpatient/ outpatient), perinatal care setting (antepartum, intrapartum, postpartum, neonatal), study population (patient group(s), sample size, special characteristics), sampling method, level of analysis (individual, population-/ group-based), source of financial support, and special considerations, e.g., study related to COVID-19 pandemic.

**Population.** We will extract the stage of pregnancy, childbirth or puerperium of included individuals and/ or study groups.

**Exposure.** We will extract reported information concerning SDoH. Since we expect high variance in reported SDoH conditions, we will summarise extracted data by clustering according to the following, predefined categories for

**Table 1. Terms for systematic literature search in PubMed according to PEO scheme.**

| | MeSH Terms | Text Words [Title/ Abstract] | |
|---|---|---|---|
| **Population** | pregnant people | obstetric | parturi* |
| | pregnancy | perinatal | peripartum |
| | peripartum period | gestation* | intrapartum |
| | postpartum period | pregnan* | intranatal |
| | | prepartum | labour/ labor |
| | | antenatal | postpartum |
| | | antepartum | postnatal |
| | | prenatal | puerper* |
| **Exposure** | social determinants of health | determinants of health | poor housing |
| | socioeconomic factors | socioeconomic/ socio-economic factor* | incarcerat* |
| | unemployment | socioeconomic/ socio-economic condition* | imprison* |
| | poverty | socioeconomic/ socio-economic status | sex worker* |
| | educational status | socioeconomic/ socio-economic characteristic* | prostitute* |
| | literacy | social condition* | discriminat* |
| | communication barriers | vulnerability | raci* |
| | health services accessibility | economic factor* | ethnic* |
| | insurance coverage | low income | race |
| | social conditions | poverty | |
| | social vulnerability | unemploy* | refugee |
| | incarceration | educational level | migrant |
| | sex workers | educational status | |
| | violence | communication barrier* | migrat* |
| | social discrimination | language barrier* | asylum |
| | racism | illitera* | food insecurit* |
| | refugees | health services accessibilit* | water insecurit* |
| | transients and migrants | access to healthcare | adolescen* |
| | food insecurity | access to health services | gender identity |
| | ethnicity | access to care | displaced |
| | war exposure | insurance status | war exposure |
| | religion | insurance coverage | religion* |
| | | housing instabilit* | religious belief* |
| | | housing insecurit* | violence |
| | | homeless* | |

*(Continued)*

**Table 1.** (Continued)

| | MeSH Terms | Text Words [Title/ Abstract] | |
|---|---|---|---|
| **Outcome** | patient safety | patient safety | medication error* |
| | patient harm | safety of care | harmful incident* |
| | medical errors | inpatient safety | safety outcome* |
| | treatment failure | unsafe care | safety incident* |
| | | unsafe healthcare | safety event* |
| | | patient harm* | adverse event* |
| | | near miss* | adverse outcome* |
| | | close call* | safety behaviour/ behavior |
| | | medical error* | perceived safety |
| | | medical mistake* | experienced safety |
| | | critical incident* | reported safety |
| | | never event* | treatment failure* |

proxy indicators of patients' SDoH: Economic stability, education access and quality, health care access and quality, neighbourhood and built environment, social and community context, individual constitutional factors, and refugee status [7,36].

**Outcomes.** We will extract any reported quantitative data and measures pertaining to perinatal patient safety. If sufficient information is reported, we will employ patient safety event taxonomy introduced by the Joint Commission on Accreditation of Healthcare Organizations and classify *types* of patient safety constraints into communication, patient management, or clinical performance, and their *causes* into system failures (organisational and technical) and human factors [24], including a category 'not classifiable' to both types and causes. In preparation to potential sub-group analyses, we will also consider classification according to data sources such as self-reported (i.e., by patients, relatives, or providers), clinical data (e.g., from clinical information systems or quality monitoring), or patient record data.

For each exposure condition and outcome, we will extract data sources, types of data, measurement tools, summary statistics, and confounders. We will also extract result data (measures of association), and effect sizes, if reported. All variables and results will be extracted as reported.

## Quality assessment

Methodological quality and risk of bias of eligible studies will be systematically assessed using the critical appraisal tools proposed by Joanna Briggs Institute (JBI), including items on appropriateness of sampling and sample size, consistency, and validity of analytical methods [41]. Each of the domains will be judged as to the possible risk of bias, and will be rated as 'high risk' and 'low risk'. If essential information concerning one or more domains is missing, risk of bias will be rated as 'unclear'. Two authors will review risk of bias independently. We will resolve potential disagreement through discussion until consensus is obtained, involving a third author. Reviewers will not be blind to the studies. Results of risk of bias assessment will be considered for sensitivity analysis of the meta-analysis results by omitting studies that are judged to be at high risk of bias.

## Data synthesis

**Primary outcome.** The review's outcome of interest will be estimates on associations between SDoH and perinatal patient safety on a study- and an individual patient level. To address expected challenges regarding variation in exposure

and outcome measures in included studies, we will also assess associations between clustered variables and outcomes (cf., data extraction). We will determine associations using correlation coefficients appropriate to each type of data.

**Association measures.** Correlation coefficients will be used as the primary measure of effect size for synthesising results across studies [35,42]. If no effect size measure has been reported in the original study, we will calculate correlation coefficients based on available quantitative data of surveyed associations. In preparation for the subsequent pooling of effect sizes, it is necessary to calculate or approximate correlation coefficients for the various reported effect sizes (e.g., odds ratios, relative risks, or standardised mean differences) using transformation formulas [42].

**Assessment of heterogeneity.** In preparation for the assessment of clinical and methodological heterogeneity, we will conduct a narrative synthesis of all included studies, using structured tables to present and compare study and patient characteristics, reported measures, and key findings [35]. Variations will be discussed with respect to their potential impact on pooling data and review findings. Subsequently, if deemed appropriate based on previous assessment of clinical and methodological heterogeneity, we will assess statistical heterogeneity by using Cochran's Q and I2 tests. Statistically significant heterogeneity will be indicated if Cochran's Q is large (to be interpreted depending on the number of included studies), or if I2 $\geq$ 50% or p < 0.1 [43]. In such cases, conducting meta-analyses of the entire set of studies may not be appropriate. Instead, the strength of individual effects will be presented visually via harvest plots. Furthermore, we will consider meta-analyses according to SDoH and outcome groups, employing the aforementioned procedures for assessing heterogeneity for each subset of studies, respectively.

**Pooling data.** If the full set of studies, or a subset of them, are sufficiently homogenous with respect to the design and outcomes (association measures, and factors controlled for), we will conduct meta-analyses using a random-effects model to calculate within- and between-study variances and pooled correlation effect sizes. Decision will be made based on assessment of heterogeneity (cf., assessment of heterogeneity). The measure for aggregated effect sizes will be pooled correlation coefficients. If appropriate, we will apply Fisher's Z transformation to normalise individual correlation coefficients [35,42].

**Subgroup and sensitivity analyses.** Subgroup analyses will be conducted to explore potential sources of between-study variability according to the following criteria: Stage of pregnancy, perinatal care settings, and countries' income-level. We will consider additional meta-regression methods, if deemed appropriate after data extraction and aggregation, and examining the intersection of cumulated exposure conditions and outcomes. To assess robustness of the meta-analysis findings, we will also perform a sensitivity analysis by omitting studies that were judged to be at high risk of bias [43].

Finally, we will present our results in a 'Summary of findings' table [44].

## Confidence appraisal

Publication bias will be assessed through visual inspection of funnel plots. We will conduct tests for asymmetry, if sufficient studies were included [45].

Additionally, we will assess quality of evidence for all outcomes using the Grading of Recommendations Development, Assessment and Evaluation (GRADE) tool. We will judge both individual and cumulative evidence as to be high, moderate, low, or very low [46]. Evidence will be assessed for all key outcomes included in meta-analyses as well as for potential additional analyses for subgroups, respectively.

## Discussion

In order to ensure safe perinatal care services, it is imperative to understand the potential risks for disadvantaged people during childbirth, the puerperium as well as for their newborns. Specifically, a profound understanding of the associations between SDoH and patient safety is essential to identify and mitigate avoidable harm – particularly, since several researchers and global health policy makers have called for the integration of equity considerations into patient safety

research [1,8,47–50]. This is of particular significance in perinatal care, where numerous studies have indicated adverse outcomes in cases of social deprivation [8–20,28]. Although previous research findings suggested that health care may contribute to existing health inequalities [15,51–54], the specific role of patient safety in the context of inequitable care remains unclear. This lack of cohesive evidence limits our understanding of risk of harm to disadvantaged patients in perinatal care. Moreover, this discrepancy poses significant challenges for advancing research methodologies, assessing care quality and safety, and developing targeted interventions. Our forthcoming systematic review aims to address this shortcoming by providing the first structured synthesis of key SDoH and patient safety measures in perinatal care. Consequently, we aim to contribute substantially to equity and safety research – within and beyond the field of perinatal care.

To date, both evidence on perinatal patient safety and the impact of SDoH is fragmented and inconclusive. In particular, this paucity of evidence can be attributed to methodological limitations regarding a clear differentiation between overall health outcomes and patient safety specific measurements. Furthermore, broad measures such as severe morbidity and mortality represent the 'tip of the iceberg' [27], and do therefore only partly contribute to a comprehensive understanding of perinatal patient safety in daily care practice [19,27], particularly given the rare occurrence of these specific events in perinatal care [55]. Thus, our consideration of a broad scope of potential safety measures according to the studies' aim and research methodologies, respectively, will facilitate a comprehensive understanding of extant patient safety research in perinatal care. Concurrently, with respect to the substantial problem of health and health care inequalities among ethnic minorities and refugees [18,21,56], race and ethnicity are investigated predominantly. Yet, other key determinants such as socio-economic status or education are underexplored [9,19,57]. This selective focus may lead to fragmented and potentially misleading conclusions regarding interactions among SDoH and perinatal patient safety. The systematic review will therefore expand and deepen knowledge through a comprehensive approach incorporating diverse data sources, exposure variables, and outcome measures, in line with established frameworks [5–7,23,24].

Despite the various advances of the review, we acknowledge potential limitations that may arise from the absence of a standardised definition or measurement framework for SDoH. This refers to the use of proxy measures, which could vary considerably between studies. Consequently, there is a risk of omitting relevant studies despite the comprehensive search strategies employed. We will mitigate this risk through citation tracing and snowballing techniques. Moreover, we expect heterogeneity in study designs, exposure, and outcome measures as well as limited data for specific subgroups of SDoH and patient safety. This heterogeneity may impede the feasibility of meta-analysis, as previously reported in similar research attempts [19,35]. Nevertheless, we aim to present the study data in the form of comprehensive descriptive summaries, narrative syntheses, and visual representation (i.e., harvest plots). This will facilitate a transparent assessment of variations between studies and enable drawing robust conclusions regarding the strength of individual effects, besides the results of meta-analyses. Lastly, multiple unmeasured confounding variables and systemic contributory factors – such as national policy regulations, health service access, minority stress, and provider implicit bias [10,15,51,52] – may limit our ability to infer direct effects or causality. In light of these challenges, the review will adhere to the Cochrane Handbook for Systematic Reviews of Interventions [58], where applicable, and to the methodological considerations for systematic reviews of associations as recommended by the JBI [35]. Future amendments to the review shall be made available for review on the study registries PROSPERO and OSF, along with the underlying rationale.

Patient safety is inherently complex, necessitating multidimensional research approaches to effectively capture the interrelationships among contributory factors [23,59,60]. However, the identification of social exposure conditions and associated safety measures is of particular importance, since a more profound comprehension of the dynamics between these factors may enhance patient safety and the quality of care during the perinatal period, with a particular emphasis on the needs of disadvantaged patients. We further acknowledge that our conceptualisation of SDoH may encourage future research into patients' social conditions such as the impact of intersectionality as well as mediating processes on perinatal patient safety. Consequently, the review's findings will contribute substantially to patient safety research, particularly by informing the design of advanced research methodologies for the study of health disparities and avoidable patient harm.

Furthermore, anticipated findings may inform development of interventions, mitigation strategies, and policies, i.e., implementation of screening tools for specific risk factors. It can thus be concluded that the results of this review provide a valuable foundation for future research through advancing understanding of SDoH and perinatal patient safety and may also empower healthcare organisations to proactively address hidden risks in the context of social deprivation that adversely affect maternal and neonatal health.

## Supporting information

**S1 Checklist. PRISMA-P checklist.** SDoH and perinatal patient safety.
(DOC)

## Author contributions

**Conceptualization:** Katharina Averdunk, Matthias Weigl.

**Methodology:** Katharina Averdunk, Céline Miani, Brigitte Strizek, Matthias Weigl.

**Supervision:** Matthias Weigl.

**Writing – original draft:** Katharina Averdunk.

**Writing – review & editing:** Céline Miani, Brigitte Strizek, Matthias Weigl.

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
