## [Editor Report · Decision Letter 0]

18 Sep 2025

Dear Dr. Averdunk,

Thank you for submitting your manuscript to PLOS ONE. After careful consideration, we feel that it has merit but does not fully meet PLOS ONE’s publication criteria as it currently stands. Therefore, we invite you to submit a revised version of the manuscript that addresses the points raised during the review process.

**ACADEMIC EDITOR:**

Include clear research questions and objectives. Follow the PICO (Patient/Population/Problem, Intervention, Comparison, and Outcome) criteria Manuscript needs major revisions.

We look forward to receiving your revised manuscript.

Kind regards,

Asma Tahir Awan, PhD (scholar). DrPH. MSHI. MPH. MBBS.

Academic Editor

PLOS ONE

Journal Requirements:

Additional Editor Comments :

The manuscript needs to be revised based on the PICO (Patient/Population/Problem, Intervention, Comparison, and Outcome) criteria. PEO (Population – Exposure – Outcome) criteria has been mentioned, but inclusion criteria of studies also mentioned intervention studies, --- which may call for a PICO (Patient/Population/Problem, Intervention, Comparison, and Outcome) criteria. Seems like more of a scoping review in the absence of a robust research question in the present state. There are no clear questions for need of research mentioned under "Objective"

---

## [Author Response · Author response to Decision Letter 1]

8 Oct 2025

08 OCTOBER 2025

Dear Dr Asma Tahir Awan, Dear Reviewers,

We thank you very much for reviewing of our manuscript entitled 'Associations of social determinants of health and patient safety in perinatal care: Protocol for a systematic review with meta-analysis’ for publication in PLOS One. We are very grateful for your thoughtful recommendations that helped us to improve the manuscript significantly.

In preparation for this resubmission, we have carefully revised our manuscript and inserted additional information in line with your recommendations. Please find below our detailed responses to your comments.

Reviewer Comment #1:

Author response:

We have revised the manuscript style and have now renamed the files in accordance with the guidelines.

Reviewer Comment #2:

Please provide a complete Data Availability Statement in the submission form, ensuring you include all necessary access information or a reason for why you are unable to make your data freely accessible. If your research concerns only data provided within your submission, please write "All data are in the manuscript and/or supporting information files" as your Data Availability Statement.

Response:

Following your suggestion, we have updated our Data Availability Statement (See ‘Additional Information’ in the submission system).

Reviewer Comment #3:

We note that the grant information you provided in the ‘Funding Information’ and ‘Financial Disclosure’ sections do not match. When you resubmit, please ensure that you provide the correct grant numbers for the awards you received for your study in the ‘Funding Information’ section.

Response:

Please acknowledge, all four authors do not receive funding for the draft of this review protocol. This information has been specified in the 'Funding Information' section. However, the Open Access Publication Fund of the University of Bonn, Germany, will provide support for open access publication. In accordance with the funding guidelines of the Open Access Publication Fund, it is necessary to include a funding statement in the manuscript concerning this matter (See 'Financial Disclosure' section). A grant number for this funding is not available.

Reviewer Comment #4:

Response: Throughout the whole manuscript, we have carefully reviewed all cited references of previously published works and thoroughly checked for relevance for inclusion.

Reviewer Comment #5:

The manuscript needs to be revised based on the PICO (Patient/Population/Problem, Intervention, Comparison, and Outcome) criteria. PEO (Population – Exposure – Outcome) criteria has been mentioned, but inclusion criteria of studies also mentioned intervention studies, --- which may call for a PICO (Patient/Population/Problem, Intervention, Comparison, and Outcome) criteria. Seems like more of a scoping review in the absence of a robust research question in the present state. There are no clear questions for need of research mentioned under "Objective".

Response: With regard to your thoughtful remark, we have decided to exclude intervention studies from the scope of our review. These studies, anyway, rarely align with the remaining inclusion criteria (p. 6, l. 120). Concurrently, after careful consideration, we have therefore decided to maintain the PEO criteria. The suitability of the Population-Exposure-Outcome scheme for reviews of association is substantiated by the well-established publication of Moola et al. (2015), providing methodological considerations of systematics review of associations. In order to provide further clarification in response to your point, we now describe our rationale for the selection of the PEO scheme (p. 5, ll. 113–114).

Additionally, following your recommendation, we have clarified the objective of the review and have amended a clear review question (p. 5, ll. 96–100).

We would further like to acknowledge your concerns regarding the distinction between a systematic and a scoping review. While the broad scope could indeed suggest a scoping approach, we chose a systematic review procedure to ensure maximum comprehensiveness and completeness of included studies. This was particularly important given the current lack of studies on patient safety in perinatal care and the heterogeneous terminology used. Our decision was supported by the work of Munn et al. (2018), who provide guidance for choosing between these two review types.

Lastly, in the course of this manuscript revision and reconsideration, the following modifications have been made during the course of processing this resubmission:

1. Exclusion criteria: For the sake of completeness, we have added case reports as an exclusion criterion (p. 6, ll. 120–121).

2. Eligibility – Outcome: We have specified the exclusion of studies that do not specifically reference to safety or harm in the context with perinatal care provision. By doing so, we aim to clarify our understanding of safety as being closely related to the service received. This approach differs from safety concerns that may arise from, e.g., poor life style or general health conditions. We believe that this specification serves clarifying the reviews’ objective and study selection (p. 7, ll. 149).

3. Databases: We had the viable opportunity to extend our literature search to additional databases such as Embase and CINAHL. On reverse, we did not perform a systematic search in Cochrane database as well as study registries due to the strong predominance of literature reviews and, for Cochrane, clinical trial registrations. We have adapted the manuscript concerning this matter (p. 2, ll. 32–33; p. 7, ll. 156–157, 'Abstract' section).

4. Translation of search terms: We have specified the information regarding the translation of search terms for the different databases, for which we have used Polyglot, provided by SR Accelerator (p. 9, l. 165).

5. Software for reviewer collaboration: We have specified in the manuscript that we use Rayyan for level 1 and level 2 screening (p. 10, l. 170).

All changes will be amended to our study registrations on OSF and PROSPERO.

We thank you in advance very much for your valuable time and consideration of our manuscript. We look forward to receiving your response. Should you have any further queries, please do not hesitate to contact us.

With kind regards, and on behalf of all my co-authors,

Katharina Averdunk M.Sc.

References cited in this letter:

Moola S, Munn Z, Sears K, Sfetcu R, Currie M, Lisy K, et al. Conducting systematic reviews of association (etiology): The Joanna Briggs Institute’s approach. JBI Evid Implement. 2015 Sep;13(3):163.

Munn Z, Peters MDJ, Stern C, Tufanaru C, McArthur A, Aromataris E. Systematic review or scoping review? Guidance for authors when choosing between a systematic or scoping review approach. BMC Med Res Methodol. 2018 Nov 19;18(1):143.

---

## [Editor Report · Decision Letter 1]

21 Oct 2025

Associations of social determinants of health and patient safety in perinatal care: Protocol for a systematic review with meta-analysis

PONE-D-25-37231R1

Dear Dr. Averdunk,

We’re pleased to inform you that your manuscript has been judged scientifically suitable for publication and will be formally accepted for publication once it meets all outstanding technical requirements.

Kind regards,

Asma Tahir Awan, PhD (scholar). DrPH. MSHI. MPH. MBBS.

Academic Editor

PLOS ONE
---

## [Editor Report · Acceptance letter]

PONE-D-25-37231R1

PLOS ONE

Dear Dr. Averdunk,

I'm pleased to inform you that your manuscript has been deemed suitable for publication in PLOS ONE. Congratulations! Your manuscript is now being handed over to our production team.

Kind regards,

on behalf of

Dr. Asma Tahir Awan

Academic Editor

PLOS ONE